# Selective Cytotoxicity of Piperine over Multidrug Resistance Leukemic Cells

**DOI:** 10.3390/molecules26040934

**Published:** 2021-02-10

**Authors:** Julia Quarti, Daianne N. M. Torres, Erika Ferreira, Raphael S. Vidal, Fabiana Casanova, Luciana B. Chiarini, Eliane Fialho, Vivian M. Rumjanek

**Affiliations:** 1Institute of Medical Biochemistry Leopoldo de Meis, Federal University of Rio de Janeiro, Rio de Janeiro 21941-902, Brazil; julia_quarti@hotmail.com (J.Q.); raphaelsilveiravidal@gmail.com (R.S.V.); 2Institute of Nutrition Josué de Castro, Federal University of Rio de Janeiro, Rio de Janeiro 21941-902, Brazil; nutri.erika.ferreira@gmail.com (E.F.); fabi_nut@yahoo.com.br (F.C.); 3Institute of Biophysics Carlos Chagas Filho, Federal University of Rio de Janeiro, Rio de Janeiro 21941-902, Brazil; daiannetorres@biof.ufrj.br (D.N.M.T.); chiarini@biof.ufrj.br (L.B.C.)

**Keywords:** chronic myeloid leukemia, multidrug resistance, piperine, collateral sensitivity, PARP-1

## Abstract

Multidrug resistance (MDR) is the main challenge in the treatment of chronic myeloid leukemia (CML), and P-glycoprotein (P-gp) overexpression is an important mechanism involved in this resistance process. However, some compounds can selectively affect MDR cells, inducing collateral sensitivity (CS), which may be dependent on P-gp. The aim of this study was to investigate the effect of piperine, a phytochemical from black pepper, on CS induction in CML MDR cells, and the mechanisms involved. The results indicate that piperine induced CS, being more cytotoxic to K562-derived MDR cells (Lucena-1 and FEPS) than to K562, the parental CML cell. CS was confirmed by analysis of cell metabolic activity and viability, cell morphology and apoptosis. P-gp was partially required for CS induction. To investigate a P-gp independent mechanism, we analyzed the possibility that poly (ADP-ribose) polymerase-1 (PARP-1) could be involved in piperine cytotoxic effects. It was previously shown that only MDR FEPS cells present a high level of 24 kDa fragment of PARP-1, which could protect these cells against cell death. In the present study, piperine was able to decrease the 24 kDa fragment of PARP-1 in MDR FEPS cells. We conclude that piperine targets selectively MDR cells, inducing CS, through a mechanism that might be dependent or not on P-gp.

## 1. Introduction

In clinical practice, cancer patients usually develop cross-resistance to drugs with different mechanisms of action and structures, making treatment inefficient. Therefore, the phenomenon known as multidrug resistance (MDR) is considered the main obstacle in the treatment of various types of cancers, including chronic myeloid leukemia (CML) [1].

There are several mechanisms related to the MDR phenotype, including overexpression of drug efflux pumps, alteration of cell cycle, inhibition of apoptosis, and induction of autophagy [2,3]. Transporters belonging to the ATP-binding cassette (ABC) superfamily, such as P-glycoprotein (P-gp), are considered the main cause of MDR. They actively transport drugs across plasma membranes, using ATP as an energy source, decreasing the intracellular concentration of those drugs [4]. In addition, cell cycle checkpoints are important targets in cancer therapy [5], because the cell cycle arrest allows DNA repair or pro-apoptotic protein induction [6]. However, even though apoptosis is an extremely well-regulated mechanism, it is quite usual for cancer cells to develop mechanisms to evade this type of death, leading to chemotherapy treatment failure [7,8]. Therefore, reactivation of apoptotic pathways in MDR cells is an important defense mechanism that can efficiently eliminate these cells [9].

One strategy to overcome MDR is the use of compounds able to selectively target MDR cells. This sensitive phenotype presented by MDR cells is known as collateral sensitivity (CS) [10,11]. Most probably, more than one mechanism is involved in this phenomenon, however they are not yet fully understood. There are some hypotheses suggesting that P-gp could confer some fragility to the MDR cell during CS induction, rather than conferring resistance [12]; for example, CS-promoting agents would be able to stimulate the P-gp ATPase activity [13], which would lead to an increase in the intracellular ATP depletion rate [14,15] and, possibly, an increase in ROS levels, which may lead to apoptosis [16]. Meanwhile, other compounds can act independently of this efflux pump [17]; for example, interfering with metabolic signaling pathways, because MDR cells are more sensitive to changes in energy metabolism than non-resistant cells [18,19].

The identification of altered signaling pathways (sensitive cells vs. MDR cells) would allow the discovery of other possible targets, besides P-gp, for CS induction. Thereby, it is necessary to consider that chemotherapy resistance mechanisms, developed during the acquisition of the MDR phenotype, appear to be complex and go far beyond P-gp overexpression. For example, in our experimental model of CML, sensitive and MDR cells differ in functional terms [20], as well as in their transcriptome [21] and proteome [22]. These K562-derived MDR cells, known as Lucena-1 and FEPS, were selected by exposure to vincristine and daunorubicin, respectively, and, after this process, they became resistant to different classes of drugs, including the most used drug in the treatment of CML, imatinib [20]. P-gp expression seems to be the main resistance mechanism of both MDR cells [20,23]. Lucena-1 cells show five-fold more copies of the *MDR1* gene (which encodes P-gp) than their K562 parental cell line [23], while FEPS cells have an even higher P-gp expression and activity compared to Lucena-1 [20]. Other differences between these cells can also be highlighted. Both MDR cell lines are resistant to UVA and H_2_O_2_ [24,25]; divide less than K562 [20]; and display changes in energy metabolism [26]. These leukemia cells also show several differences in the expression of genes related to cell death, among others [21].

Previous data from our research group also indicate remarkable differences in the levels of poly (ADP-ribose) polymerase-1 (PARP-1) enzyme between the drug-sensitive leukemia cell line (K562) and the most resistant leukemia cell line of our experimental model (FEPS), wherein only FEPS cells have detectable levels of the 24 kDa fragment of PARP-1 [26]. Other workers showed that this fragment binds to DNA, competing with the repair enzyme PARP-1, attenuating PARP-1 overactivation [27,28] and saving cells from energy depletion (NAD+ and ATP) and cell death by necrosis [29]. Therefore, the attenuation of PARP-1 activity in MDR FEPS cells might be an important protection mechanism against cell death contributing to the resistant phenotype of this cell line, because it has high energy demand due to the expression and activity of efflux pumps, such as P-gp [26]. Therefore, PARP-1 could be an important target against MDR.

Phytochemicals, also known as bioactive food compounds, can target different signaling pathways in cancer cells, especially those related to cell cycle arrest, apoptosis induction, autophagy modulation, and reversing drug resistance [30,31,32]. Among these compounds, piperine, a phytochemical from black pepper, can be highlighted due to its anti-cancer and anti-MDR activities [33].

Piperine is able to modulate the MDR phenotype in some experimental models, such as breast, lung, colon and lymphoma cancer. This compound inhibited the gene expression and activity of the P-gp and BCRP efflux pumps (in breast cancer cells) and the MRP-1 efflux pump (in lung cancer cells) [34]. The action of this phytochemical on colon cancer and lymphoma cells with MDR phenotype has also been demonstrated, by enhancing the cytotoxic effect of the chemotherapeutic drugs doxorubicin and mitoxantrone, respectively. In addition to having inhibited P-gp activity in these two types of cancers [35], a study by Morsy et al. (2018) [36] found, through in silico and in vitro studies, that piperine was able to inhibit P-gp in ovarian cancer cells, suggesting that this phytochemical would be a promising adjuvant in the treatment with doxorubicin.

Although previous results have demonstrated piperine anti-cancer activity, in preclinical studies [37], its effect on CS promotion has yet not been thoroughly investigated.

Thus, the aim of this study was to investigate the effect of piperine on CS induction in CML MDR cells, as well as the mechanisms involved, including its dependence or not on P-gp expression.

## 2. Results

### 2.1. Piperine Was More Effective to MDR Leukemic Cell Lines Than to Parental Cells

The first step was to analyze the effect of piperine on the metabolic activity and viability of the parental cell line, K562, and the MDR cell lines, Lucena-1 and FEPS. After treatment with different concentrations of piperine for 48 (Figure 1A), 72 (Figure 1B) and 96 h (Figure 1C), analysis by 3-(4,5-dimethylthiazol-2-yl)-2,5-diphenyl-tetrazolium bromide (MTT) reduction assays were performed in order to evaluate cellular metabolic activity. A viability test with trypan blue was also performed after 72 h of piperine treatment (50 or 100 µM) (Figure 1D).

According to the results obtained and shown in Figure 1, it was observed that piperine reduced the metabolism and cell viability of K562, Lucena-1 and FEPS cells in a dose-dependent manner. Besides that, in general, piperine was shown to produce a greater impact to the MDR cell lines, Lucena-1 and FEPS, than to the parental one, K562, suggesting the occurrence of the CS phenomenon. For example, after 96 h of piperine treatment, a concentration of 150 μM was necessary to produce a significant alteration (*p* < 0.05, ANOVA with Tukey’s post-test) in the metabolic activity of K562 cells, whereas with Lucena-1 and FEPS cells the concentration necessary was only 75 μM and 25 μM, respectively. When comparing IC_50_ values, it was also found that piperine was more cytotoxic to the FEPS cell line than to Lucena-1 (*p* < 0.05, ANOVA with Tukey’s post-test) after treatment for 72 and 96 h. Besides that, cell viability analysis with trypan blue also showed that FEPS cells were more affected by piperine compared to the other two leukemic cell lines (Figure 1D).

### 2.2. Piperine Induced Morphological Changes to a Greater Degree in FEPS Cells

Cells treated with 100 µM of piperine for 72 h were stained with Quick Panoptic in order to examine cell morphology (Figure 2A). Bright field microscopy images indicated that piperine caused significant morphological changes in K562, Lucena-1, and even more markedly in FEPS cells. An increase in vacuole formation was observed in the cytoplasm of FEPS cells compared to the other cell lines (Figure 2A).

The forward scatter (FSC-H) and side scatter (SSC-H) parameters were analyzed by flow cytometry in order to confirm the morphological changes induced by piperine (Figure 2B), with FSC-H indicating cell size and SSC-H showing the complexity or granularity of the cell. Leukemic cells were affected by piperine treatment, according to size and granularity parameters. Moreover, piperine, at a concentration of 50 µM, was enough to cause morphological changes on FEPS cells, whereas in K562 and Lucena-1 cells this effect was only more evident at 100 µM concentration (Figure 2B).

The altered parameter in this analysis was granularity, which can be observed by the accumulation of cells in the region 1 (R1) of the graphs. MDR FEPS cells presented the highest percentage of cells with this characteristic, both after treatment with 50 µM (11-fold increase) and 100 µM of piperine (18-fold increase) (Figure 2C). However, the percentage of cells with changes in size and granularity (R2), concomitantly, was similar among the studied strains (Figure 2C). In addition, none of the leukemic cells showed changes in their size (R4) after treatment with piperine.

Therefore, these data are in accordance with what was observed in cytology; both suggest that the effect of piperine is more significant in MDR FEPS cells. In addition, the accumulation of cytoplasmic vacuoles could be reflecting the increase in granularity mainly observed in piperine-treated FEPS cells.

### 2.3. Piperine Did Not Alter Cell Cycle Distribution, but Induced Apoptosis of MDR Cells

Next, experiments were carried out to verify the effect of piperine on the cell cycle distribution of the K562, Lucena-1 and FEPS cell lines using flow cytometry (Figure 3).

The concentrations of piperine used, as well as the exposure time, were not sufficient to significantly arrest the cell cycle of the studied leukemic cell lines at some specific phase (Figure 3).

In addition to the investigation of cell cycle, we analyzed DNA fragmentation, as an indicative of cell death. For this, the quantification of cells in the Sub-G1 phase was evaluated (Figure 3). Piperine treatment with 100 μM only increased Sub-G1 phase of the MDR cells, Lucena-1 and FEPS, indicating DNA fragmentation (≅15% and 20%, respectively). However, a lower concentration of piperine (50 μM) was already capable of rising the Sub-G1 phase of FEPS cells (≅15%). Besides that, a modest decrease in the percentage of cells in G2/M phase was also observed in the three studied cell lines (≅5–10% reduction), after treatment with 50 (FEPS) or 100 µM piperine (K562 and Lucena-1), probably due to the increase (or trend to increase, in the case of K562) in the Sub-G1 phase (Figure 3).

In order to identify the effect of piperine upon cell death of leukemic cell lines, experiments with annexin-V (stains apoptotic cells) and propidium iodide (PI) (stains necrotic cells) were performed (Figure 4).

According to Figure 4, piperine only increased (*p* < 0.05) the apoptosis rate (apoptosis plus late apoptosis) of K562 cells after 100 µM piperine. Meanwhile, 50 µM piperine concentration was already able to increase (*p <* 0.05) the apoptosis rate of the two MDR cells, Lucena-1 (increase of 1.6-fold in relation to control) and FEPS (increase of 1.8-fold in relation to control). However, the concentration of 100 µM piperine generated an even more pronounced effect on these cell lines (*p <* 0.01), about 3.6- and 4.5-fold in relation to control for Lucena-1 and FEPS, respectively, corroborating previous experiments.

When etoposide was used as a positive control, only ≅25% of K562 and Lucena-1 cells remained negative for annexin-V and PI. Despite the sensitivity to piperine, the staining profile produced by etoposide indicated that FEPS cells were resistant to this chemotherapeutic drug (Figure 4M).

Subsequently, we investigated, by immunofluorescence, whether piperine-induced apoptosis was associated to caspase-3 activation (Figure 5). 

Cleaved caspase-3 was found in MDR cells (Lucena-1 and FEPS) treated with 100 µM of piperine for 72 h, but not in K562 cells (Figure 5). These results suggest that proteolytic activation of caspase-3 induced by piperine in leukemic cells occurs only in cells with the MDR phenotype. In addition, the immunofluorescence for cleaved caspase-3 was more intense in piperine-treated FEPS cells, corroborating the previous experiments (Figure 5).

### 2.4. Piperine Partially Reverses the Resistance and Shows Synergism to Vincristine on Lucena-1 Cells

Experiments were carried out to test whether piperine would also be able to reverse the resistance of Lucena-1 and FEPS cells to the chemotherapeutic agent vincristine. The effect of the association of piperine with vincristine on cell metabolic activity of Lucena-1 and FEPS cells was analyzed by the MTT reduction assay. For this, cells were treated with 400 nM of vincristine in the presence or absence of piperine (5 μM or 50 μM) or verapamil (5 μM), as a positive control. After treatment with the compounds for 72 h, analyses were performed (Figure 6).

To test the effect of piperine in association with vincristine, two different concentrations of piperine were chosen based on the MTT reduction assay shown in Figure 1B: a non-toxic concentration for MDR cells (5 µM), capable of identifying a possible reversal effect of resistance to vincristine; and an intermediate concentration (50 µM), related to CS induction of MDR cells, to investigate a possible synergistic effect between these compounds (Figure 6).

Only in Lucena-1 cells did the addition of 5 µM piperine to vincristine treatment cause a significant reduction in the cell metabolic activity, compared to the treatment with the isolated vincristine. However, the addition of 50 µM piperine to vincristine treatment caused an increased MTT reduction in both Lucena-1 and FEPS cells (Figure 6). 

It is worth mentioning that in FEPS cells, the effect of 50 µM piperine per se (≅60% reduction in cell metabolic activity) or 50 µM piperine associated with vincristine (≅75% reduction in cell metabolic activity) was quite similar (no statistical difference by ANOVA with Tukey’s post-test), so it appears to be an additive effect with vincristine (≅10% reduction in cell metabolic activity), and not a synergistic effect. While for the Lucena-1 cell line, 5 µM piperine partially reversed resistance to vincristine and the 50 µM concentration produced an increased MTT reduction of ≅25% compared to the sum of the effects of the isolated compounds, suggesting a synergistic effect with vincristine (Figure 6). 

These data indicate that, when associated with the chemotherapeutic agent vincristine, piperine only affects the Lucena-1 cell line, partially reversing the resistance and presenting a synergistic effect. However, the effect of verapamil, a well-known inhibitor of P-gp activity, and an inducer of CS in some MDR cells [13], was more pronounced than that of piperine (ANOVA with Tukey’s post-test), regarding the reversal effect to vincristine (Figure 6).

### 2.5. P-gp Expression Is Partially Necessary for Piperine’s Cytotoxic Effect in MDR Cells

Previous experiments pointed to a more pronounced effect of piperine on P-gp-expressing cell lines, Lucena-1 and FEPS. In order to test the P-gp involvement in piperine’s cytotoxic effect, experiments were carried out with Lucena-1 and FEPS cells that were previously silenced for the *MDR1* gene [20]. Briefly, cells were transfected with *MDR1*-shRNA and mock plasmids. Cells transfected with *MDR1*-shRNA plasmids that were negative for *MDR1* expression were selected (*MDR1* shRNA). For the cells transfected with mock plasmid, those expressing *MDR1* were selected (MOCK shRNA).

Silenced cells were incubated with different concentrations of piperine for 72 or 96 h and the MTT reduction assay was performed (Figure 7).

Piperine was slightly less cytotoxic (*p* < 0.05) to silenced cells (Lucena-1 and FEPS) at the times studied (Figure 7). This result suggests that P-gp expression might be partially necessary for pipeline’s cytotoxic effect. However, silenced cells were still affected by piperine, indicating the occurrence of other mechanisms concomitantly.

### 2.6. Piperine Does Not Seem to Be a Direct Inhibitor nor a Competitive Substrate of P-gp

Because P-gp expression seemed to be, at least in part, a prerequisite for the enhanced toxicity of piperine, we evaluated whether this compound could be altering the expression of this protein, using an anti-P-gp antibody (Figure 8).

The data demonstrated that the treatment of MDR cells with the highest piperine concentration (100 μM) for 72 h induced P-gp expression in both Lucena-1 and FEPS cells (Figure 8).

A direct effect of piperine on P-gp transport activity was also studied. For this, P-gp-related transport activity measurements using the fluorescent probe Rhodamine-123 (Rho-123) were conducted (Figure 9).

Piperine did not alter the P-gp mediated Rho-123 efflux of MDR cells, contrary to what is seen when a classic inhibitor of P-gp activity such as verapamil is used (Figure 9). The fact that the addition of piperine did not lead to Rho-123 accumulation inside the cells suggests that piperine is not a direct inhibitor nor a competitive substrate of this efflux pump.

### 2.7. Piperine Downregulates PARP-1 Protein Levels and PARP-1 24 kDa Fragment in MDR FEPS Cells

We have previously found that only the most resistant cell line, FEPS, has detectable levels of the 24 kDa PARP-1 fragment [26]. In order to investigate P-gp independent mechanisms, we tested whether piperine could modulate this enzyme (Figure 10).

Similar levels of PARP-1 (116 kDa) were observed between K562 (0.77 AU), Lucena-1 (0.89 AU) and FEPS cells (0.66 AU). However, only MDR FEPS cells showed 24 kDa fragment expression (0.81 AU) (Figure 10).

Regarding the effect of piperine, the most significant reduction in PARP-1 (116 kDa) protein levels was observed in MDR FEPS cells (2.2-fold, compared to control). In addition, this compound further reduced the amount of the 24 kDa fragment of PARP-1 in this same cell line (2.8-fold, compared to control). In K562 cells, piperine had an effect on reducing PARP-1 content (89 kDa) (2.8-fold, compared to control) (Figure 10). Thus, piperine can modulate PARP-1, with these changes being more noticeable in the drug-sensitive (K562) and in the most resistant cell line of this experimental model (FEPS), despite having distinct effects and/or intensities. 

## 3. Discussion

In this study, the effect of piperine on sensitive (K562) and drug-resistant leukemic cells (Lucena-1 and FEPS) was investigated. Overall, our results indicated that this phytochemical showed a more pronounced effect on leukemic MDR cells. The importance of the present paper relies on the novelty showing this property of piperine in inducing CS that has not been described for any other cancer experimental model.

More than 200 studies have been published on the effects of piperine, or its analogs, against cancer, indicating an important chemopreventive and chemotherapeutic action in both in vitro and in vivo analysis [33,38,39]. However, to date, no clinical study with cancer patients has been published evaluating piperine safety or testing its anticancer effects. There is only one phase I clinical trial in progress that aims to evaluate, in cancer patients, the side effects and best dose of curcumin, a phytochemical present in turmeric, combined with piperine in reducing inflammation for ureteral stent-induced symptoms [40].

Although clinical pharmacokinetic studies of piperine are scarce, there are indications that oral piperine supplementation [41] or perhaps the intake of black pepper itself [42] would lead to micromolar concentration ranges of piperine in the blood. Furthermore, preliminary studies are encouraging and have shown to improve piperine’s bioavailability through nanoformulations and encapsulation in lipid bodies [43,44]. Another possibility is the use of piperine analogs in order to improve the efficiency of this compound, and, consequently, reduce its concentrations [39]. Therefore, contemporary advancements increasingly contribute to the possibility, in the near future, of using piperine, or its analogs, as a drug.

In vitro experiments conducted by other authors showed that piperine has a reduced toxic effect on normal human cells such as osteoblasts, fibroblasts and mammary cells when compared to cancer cells [45,46,47]. Chuchawankul et al. (2012) [48] also evaluated the effect of piperine on cells of the human immune system (peripheral blood mononuclear cells—PBMC) and demonstrated that piperine was not toxic for these cells, obtaining IC_50_ of approximately 350 μM (8.5-fold higher than that found for FEPS cells in the present study). This evidence suggest that this compound might cause few side effects, however clinical studies are needed to confirm this hypothesis. In order to understand the effects related to CS induction, experiments were performed to evaluate the effect of piperine on cell metabolism, viability, morphology, cell cycle progression and cell death.

Using the MTT assay, which measures the metabolic activity of the cell line, it was possible to observe that piperine toxicity differed among the MDR cells, being more expressive on FEPS compared to Lucena-1 cells. When a viability test was used, such as trypan blue exclusion, FEPS was also the cell line with the lowest percentage of living cells after treatment with piperine. In contrast, Daflon-Yunes et al. (2013) [20] demonstrated that FEPS cells are the most resistant ones of this experimental model; it is necessary to use higher concentrations of chemotherapeutic drugs to produce the same effect observed in Lucena-1 cells.

Observing cell morphology after treatment with piperine, differences were found between the three leukemic cell lines. Piperine-treated FEPS showed the highest formation of vacuoles in the cytoplasm. The increased granularity may have been caused by stress-induced injury to a greater degree in piperine-treated FEPS.

It was also considered that the pronounced effect of piperine on MDR cells, Lucena-1 and FEPS, could reflect a cell cycle arrest. However, this was not the case. Unlike our results, the work by Han et al. (2008) [49] using P-gp expressing cell lines (human colon cancer cells, that naturally express the *MDR1* gene, and porcine kidney cells stably transfected with the human P-gp) and piperine (50 or 100 μM for 48 h), observed a small, but significant, arrest at the G0/G1 phase of the cell cycle.

Furthermore, experiments to identify cell death-related proteins showed that piperine induces cleavage of caspase-3 in MDR Lucena-1 and FEPS cells, indicating apoptosis induction. Although it was not possible to find reports on the induction of apoptosis by piperine by itself in MDR cells, piperine (concentration range of 50–200 µM) promoted apoptosis and caspase-3 cleavage in other experimental cancer models, such as melanoma [50,51], oral squamous cell carcinoma [52], human cervical adenocarcinoma [53], and gastric cancer [54].

Other authors also described, in a human leukemic cell line (HL-60), that piperine could produce morphological changes, such as increased vacuoli nuclear fragmentation, as well as increased annexin-V staining, and cell cycle arrest at the S-phase of the cell cycle [55]. These data reinforce some of the results presented in this paper and suggest that piperine is a compound with important anticancer properties.

In addition to CS induction, we tested whether piperine could reverse the resistance of Lucena-1 and FEPS cells to the chemotherapeutic agent vincristine. It could be observed that piperine increased the sensitivity of Lucena-1 cells to vincristine. These data are consistent with other reports. Li et al. (2011) [34] showed that concentrations of 12.5, 25, and 50 µM of piperine reversed doxorubicin and mitoxantrone resistance of breast cancer MCF7/DOX cells and doxorubicin resistance of lung cancer A-549/DDP cells. In addition, piperine (25 and 50 μM) potentiated doxorubicin-induced cytotoxicity on ovarian NCI/ADR-RES cells [36]. Piperine can also significantly decrease the IC_50_ value of doxorubicin in Caco-2 cells and CEM/ADR 5000 cells, thus showing chemosensitizing activity at concentrations between 25 and 65 µM [35]. However, in the present study, piperine had no additional effect with vincristine on FEPS cells. These data suggest that, in the more resistant cells, the CS-inducing ability of piperine is more expressive than its reversal or synergism effect. Moreover, Syed et al. (2017) [39] also showed that a low concentration of piperine, 2 µM, was not able to reverse the resistance to vincristine, colchicine or paclitaxel of drug resistant KB (cervical) and SW480 (colon) cancer cells.

Regarding the mechanism of CS induction, it has been described that numerous CS agents, such as tiopronin, austocystin D, NSC73306, Dp44mT and desmosdumotin B, show effectiveness proportional to the amount of P-gp expressed, with cells expressing high levels of P-gp showing remarkable CS, and cells with low levels of P-gp being more tolerant to the CS agent [17,56,57]. In this context, piperine cytotoxicity was more highlighted in the Lucena-1 cell line than K562, with Lucena-1 showing five-fold more copies of the *MDR1* gene than its K562 parental cell line [23]. In addition, piperine was even more cytotoxic to the FEPS cell line, whose P-gp expression is at least two-fold higher than in Lucena-1 cells as described before [23]. These data suggest that the mechanism of piperine in inducing CS could be dependent on the presence of P-gp on the membrane of such tumor cells.

Therefore, we used silenced cells for the *MDR1* gene. These cells were previously established in our laboratory and purified by cell sorting, with the silencing efficiency estimated at 80.76% for Lucena-1 and 85.23% for FEPS [20]. The results indicated that the downregulation of P-gp partially reverses their sensitivity to piperine, demonstrating a possible correlation between P-gp expression and piperine effect. A similar situation has been observed using the multidrug resistant CH^R^C5 ovarian cells, where down regulation of P-gp completely reversed CS to verapamil [13]. However, even though a decrease in sensitivity could be observed in silenced cells in the present study, these cells continued to be affected by piperine, indicating the presence of other complementary mechanisms. Daflon-Yunes et al. in 2013 [20] characterized the MDR cell lines that were used in the present work and found that despite P-gp expression possibly being the main cause of the MDR phenotype of these cell lines, other mechanisms are also involved, because P-gp silencing only partially reversed the resistance of the cells to the chemotherapeutics used in that study.

To date, only one study has evaluated the effect of piperine on non-solid tumors which overexpress P-gp (human T-cell lymphoma) [35]. However, these authors did not observe any effect on CS induction. These data, together with our findings, indicate that, probably, the overexpression of P-gp by itself is not enough for CS induction by piperine.

In sequence, we evaluated whether piperine could directly interact with P-gp. Experiments measuring P-gp expression and drug efflux were done to evaluate this possibility. After 72 h treatment with piperine, P-gp expression was increased in both Lucena-1 and FEPS cell lines. This agrees with what has been described by Han et al. (2008) [49] using a human colon cancer cell line which overexpressed the human P-gp (Caco-2), where the authors verified, after 72 h of piperine treatment (50 and 100 μM), an increase in P-gp mRNA and protein expression. 

Our group has investigated the importance of P-gp expression for the development of the MDR phenotype through the analysis of the proteomic profile of Lucena-1 and FEPS silenced for *MDR1*. It was demonstrated that this silencing caused several changes in proteins of other signaling pathways, suggesting an interconnection between the different resistance mechanisms [22]. 

Contrary to what has been described by other authors in MDR cancer cells [34,35] the function of this efflux pump was not affected by piperine in our model. The main difference is that our protocol is more reliable to evaluate the possibility that piperine could directly interact with P-gp, as a competitive inhibitor, while other studies were more interested in investigating whether pre-incubation with piperine for a few hours would modulate P-gp activity. Our data indicate that this phytochemical does not seem to be a competitive inhibitor of P-gp. Morsy et al. (2018) [36] showed that piperine has binding affinity to P-gp, in assays in silico, but they also suggested that piperine only possess a non-competitive inhibitory effect and is possibly not a substrate of P-gp in ovarian MDR cell lines, even though these authors admit that these last conclusions still need to be confirmed through other experiments. Similar to our findings, other CS agents (NSC73306 and KP772) show indirect P-gp-mediated CS; they demonstrate increased toxicity to P-gp-expressing cells, but are not P-gp substrates, nor inhibit P-gp function [58,59].

In addition to P-gp, we suggest, in the present study, another possible target for the CS induction—PARP-1 modulation. Caspase-3 may be responsible for PARP-1 cleavage in cancer cells, resulting in the increase in 89 kDa and 24 kDa fragments [28,60,61]. However, although piperine reduced 116 kDa PARP-1 levels in FEPS cells, there was no concomitant increase in the 89 kDa and 24 kDa fragments, suggesting that PARP-1 cleavage would not be correlated with caspase-3 activation following piperine treatment in FEPS cells. In fact, piperine decreased the 24 kDa PARP-1 fragment in FEPS cells, indicating that other PARP-1 fragments, which could not be identified in the present study, were likely generated. There are also other proteases such as granzyme and matrix metalloproteinases that may be responsible for the formation of PARP-1 fragments [28]. However, to the best of our knowledge, this is the first time that the decrease in 116 kDa PARP-1 protein levels concomitant with a reduced 24 kDa fragment of PARP-1 has been observed; thus, piperine could be modulating some unknown protease able to cleave PARP-1 into other fragments or to induce PARP-1 degradation.

MDR FEPS were the only cells that presented detectable amounts of 24 kDa fragment of PARP-1, suggesting the development of a mechanism capable of preventing PARP-1 overactivation, which could protect these cells against cell death. However, to some extent, piperine impairs this survival mechanism, reducing 116 kDa PARP-1 protein levels and its 24 kDa fragment, indicating the possibility that PARP-1 could be involved in piperine’s cytotoxic effect. However, further studies are necessary to analyze whether the CS phenomenon observed is related to PARP fragmentation or degradation.

In summary, the CML cell lines used in the present study responded differently to the action of piperine. Instead of MDR cells showing resistance to the action of piperine, they showed greater sensitivity to this compound, characterizing, for the first time, the property of this phytochemical in inducing CS (Figure 11).

The MDR phenotype is a very complex phenomenon that has been extensively studied for the discovery of new therapeutic strategies. CS promoters, such as piperine, possibly act through different signaling pathways, being dependent or not on P-gp expression, in order to bypass the different mechanisms related to the MDR phenotype. The present work identified that the toxicity of piperine was partially proportional to the P-gp expression of MDR cells, and we also speculated that the differential expression of PARP-1 (especially the 24 kDa fragment) between the drug-sensitive cell line, K562, and the more resistant cell line, FEPS, may have enabled the selective toxic effect of piperine on FEPS cells (Figure 11). Identification of CS-promoting agents, as well as their mechanisms of action, may lead to the discovery of compounds that are effective in preventing MDR or making chemotherapeutic treatment efficient again by selectively killing MDR cells.

## 4. Materials and Methods

### 4.1. Cell Culture

The human erythroleukemic cell line K562, from a CML patient in blast crisis [62], and its counterparts, Lucena-1 and FEPS, were all maintained in RPMI-1640 medium, pH 7.4, supplemented with 50 mmol/L β-mercaptoethanol, 25 mmol/L HEPES, 60 mg/L penicillin, 100 mg/L streptomycin (all items obtained from Sigma Chemical Co., St Louis, MO, USA) and 10% fetal bovine serum (FBS) from Cultilab (São Paulo, Brazil). Lucena-1 and FEPS cells were developed in our laboratory by continuous exposure of K562 cells to increasing concentrations of cytotoxic drugs, vincristine sulfate (Sigma Chemical Co., St. Louis, MO, USA), for Lucena-1, and daunorubicin hydrochloride (Sigma Chemical Co., St. Louis, MO, USA), for FEPS, as described by Rumjanek et al. (2001) [23] and Daflon-Yunes et al. (2013) [20]. 

Lucena-1 and FEPS cells were maintained with 60 nM of vincristine sulfate and 500 nM of daunorubicin hydrochloride, respectively. To perform the experiments, the drugs were removed from the cell culture for three days. All cells were passaged at a concentration of 2 × 10^4^ cells/mL every three or four days and kept at 37 °C in a 5% CO_2_ humidified environment. 

Lucena-1 and FEPS cells silenced for the *MDR1* gene expression were previously established using transfection with shRNA plasmids [20]. Briefly, cells were transfected with *MDR1*-shRNA and mock plasmids. Cells transfected with *MDR1*-shRNA plasmid that were negative for *MDR1* expression were selected against those that had any amount of *MDR1* expression. For cells transfected with mock plasmid, those expressing *MDR1* were selected. Geneticin (Sigma Chemical Co., St. Louis, MO, USA) was added for the selection of successfully transfected cells. Transfection efficiency was monitored by flow cytometry, and, after 2 weeks, to guarantee a phenotypic homogeneous cell culture, both strains were subjected to cell sorting via MoFlo (Beckman Coulter^TM^, Fullerton, La Habra, CA, USA).

### 4.2. MTT Assay with Human Leukemic Cells

Leukemic cells (2 × 10^4^ cells/mL) were incubated for 48, 72 and 96 h with piperine (2.5–300 µM) (Sigma Chemical Co., St. Louis, MO, USA) diluted in 0.25% dimethyl sulfoxide (DMSO) (Sigma Chemical Co., St. Louis, MO, USA). Cell metabolic activity was assessed by MTT (Sigma Chemical Co., St. Louis, MO, USA) colorimetric assay [63]. For this, 20 µL of MTT, diluted in phosphate-buffered saline (PBS), were added at a final concentration of 0.5 mg/mL. The cells were then kept at 37 °C, 5% CO_2_ for 3 h. After centrifugation, 200 µL of DMSO were added in order to dissolve the dark blue crystals formed by the reduction in MTT. The absorbance of the converted dye in living cells was measured at a wavelength of 492 nm. The IC_50_ values were calculated from dose–response curves; the IC_50_ was defined as the concentration of drugs that reduced the cell metabolic activity to 50% of the control. GraphPad Software 5.0 was used for the IC_50_ calculations.

### 4.3. Trypan Blue Assay with Human Leukemic Cells

Leukemic cells (2 × 10^4^ cells/mL) were incubated for 72 h with piperine (50 or 100 µM). Cell viability was assessed by trypan blue dye (Sigma Chemical Co., St. Louis, MO, USA). The cell suspension (10 μL) was transferred to eppendorfs, then 10 μL of 0.4% trypan blue solution was added. After homogenization, the cells were counted using a Neubauer chamber [64].

### 4.4. Quick Panoptic Stain

In order to identify possible changes in leukemic cell morphology, cytospin slides were prepared using a Cytospin centrifuge (Thermo Fisher Scientific, Waltham, MA, USA). Leukemic cells (2 × 10^4^ cells/mL) were treated with 100 µM of piperine for 72 h. Subsequently, Quick Panoptic Staining was performed using a commercial hematological kit (Laborclin^®^ produtos para laboratórios Ltd.a., Pinhais, PR, Brazil). Following the manufacturer’s recommendations, each slide was subjected to a fixative action (0.1% triarylmethane solution) for 1 min. Then, the slides were sequentially immersed in two dye solutions (0.1% xanthene solution and 0.1% thiazine solution) for 1 min each. After the last immersion, the slides were washed in distilled water and air-dried prior to evaluation under a bright field microscopy (OLYMPUS IX81). Images were obtained using the Olympus cellSens Standard program.

### 4.5. Cell Cycle 

Cell cycle distribution was analyzed by flow cytometry as described previously [65]. Leukemic cells (2 × 10^4^ cells/mL) were treated with 50 or 100 µM of piperine for 72 h. The cells (1 × 10^5^ cells) were harvested, washed twice with PBS, and incubated with 0.3 mL of PBS containing 1 mg/mL RNase A, 0.1% Triton X-100, and 50 µg/mL PI (Sigma Chemical Co., St. Louis, MO, USA) for 10 min at room temperature in the dark. The stained cells were analyzed using a FACScalibur laser flow cytometer equipped with Cell Quest software (Becton Dickinson, San Jose, CA, USA). The percentage of cells in the sub-G1, G1, S and G2/M phases was quantified using the Summit v4.3^®^ (DAKO) Software.

### 4.6. Apoptosis

The assays were performed using the annexin V/PI Apoptosis Assay kit (eBioscience, San Diego, CA, USA) according to the manufacturer’s instructions. Leukemic cells (2 × 10^4^ cells/mL) were treated with 50 or 100 µM of piperine for 72 h. Cells were harvested, washed with PBS, and resuspended using manufacturer’s binding buffer. Subsequently, cells were incubated with binding buffer containing annexin-V conjugated to fluorescein isothiocyanate (FITC) for 20 min at room temperature and protected from light. After incubation, PI was added to each point at the time of reading at the flow cytometer. The stained cells were analyzed using a FACScalibur laser flow cytometer equipped with Cell Quest software. The percentage of cells stained with PI and/or annexin-V was quantified using the Summit v 4.3^®^ (DAKO) Software.

### 4.7. Immunofluorescence

Leukemic cells (2 × 10^4^ cells/mL) were treated with 100 µM of piperine for 72 h. Cytospin centrifugation (Thermo Fisher Scientific) was performed and the cells were fixed with 4% paraformaldehyde and washed in PBS before immunofluorescence staining. After that, the cells were permeabilized with 0.5% Triton X-100. The cells were then incubated with 1% bovine serum albumin (BSA) in PBS for 30 min at room temperature and incubated with anti-rabbit cleaved caspase-3 (1:400; Cell Signaling; #9661) in 1% BSA overnight at 37 °C. Following washes with PBS, cells were incubated for 2 h at room temperature with Alexa Fluor 555-conjugated donkey anti-rabbit antibody (Thermo Fisher Scientific) diluted in 1% BSA (1:1000), followed by washes in PBS. After that, cells were incubated with DAPI solution (1 mg/mL, Sigma Chemical Co., St. Louis, MO, USA) for 5 min, for nuclear staining. The cells were then washed in PBS and mounted with *para*-phenylenediamine (PPD). Slides were examined in three distinct fields for each slide in a fluorescence microscope (Spinning Disk) and the images were analyzed using the program Zen Pro 2012 (blue edition).

### 4.8. P-gp Expression 

The detection of P-gp cells surface was measured using FITC mouse anti-human P-gp clone 17F9 (Becton, Dickinson and Company, Franklin Lakes, NJ, USA) antibody against external epitopes of this protein [20]. In brief, MDR leukemic cells (2 × 10^4^ cells/mL) were treated with 50 or 100 µM of piperine for 72 h. The cells (2 × 10^5^) were washed in PBS with 5% FCS and then incubated with 10 μL of the antibody (1:5 dilution) for 30 min at 4 °C. After that, cells were washed in PBS with 5% FCS and fluorescence was assessed using a FACScalibur laser flow cytometer equipped with Cell Quest software. The data were analyzed using the Summit v 4.3^®^ (DAKO) Software.

### 4.9. P-gp Activity 

P-gp-related transport activity was investigated using the Rho-123 fluorescent probes (Sigma Chemical Co., St. Louis, MO, USA) in an efflux assay [20]. In brief, MDR leukemic cells (1 × 10^5^ cells) in RPMI medium with 10% FCS were incubated with Rho-123 (final concentration of 200 ng/mL for Lucena-1 or 500 ng/mL for FEPS) for 30 min at 37 °C in an atmosphere of 5% CO_2_. This incubation was performed in the presence or absence of piperine (50 or 100 µM) and positive control, verapamil (5 µM) (Sigma Chemical Co., St. Louis, MO, USA). After this time, cells were washed and re-incubated with or without piperine or verapamil for a further 15 min in medium, in the absence of the dye, to allow its extrusion. Cells were then resuspended with cold PBS and dye accumulation was assessed immediately using a FACScalibur laser flow cytometer equipped with Cell Quest software. The data were analyzed using the Summit v 4.3^®^ (DAKO) Software. The different concentrations of Rho-123 allowed perceiving the difference of Lucena-1 and FEPS P-gp activity.

### 4.10. Preparation of Cell Lysates

Leukemic cells (2 × 10^4^ cells/mL) were treated with 100 µM of piperine. After treatment for 72 h, cells were washed with PBS and lysed in liquid nitrogen. The cells were then resuspended in ice-cold buffer (5 mM Tris-HCl, 10 mM ethylenediamine tetraacetic acid, 5 mM sodium fluoride, 1 mM sodium orthovanadate, 1 mM phenylarsine oxide, 1 µM okadaic acid and 1 mM phenylmethylsulfonyl fluoride (pH 7.4)) with a freshly added protease inhibitor cocktail (1.04 mM 4-2-aminoethylbenzenesulfonyl fluoride, 15 µM pepstatin A, 14 µM E-64, 40 µM bestatin, 20 µM leupeptin, and 0.8 µM aprotinin). The lysate was collected, sonicated, and cleared by centrifugation at 8000 rpm for 5 min at 4 °C; the supernatant (total cell lysate) was then collected, aliquoted, and stored at −80 °C. The protein concentration was determined according to Lowry’s method [66]; BSA was used as standard.

### 4.11. Western Blot Analysis

Equal amounts of total cellular proteins (100 μg) were resolved by sodium dodecyl sulphate–polyacrylamide gel electrophoresis (SDS-PAGE) and transferred onto polyvinylidene difluoride (PVDF) membranes (Immobilon P, Millipore, Bedford, MA). Membranes were blocked overnight at 4 °C in Tris-buffered saline containing 1% Tween 20 (TBS-T) and 5% BSA and incubated for 2 h with the primary antibody (1:1000). The membranes were then washed with TBS-T and incubated with a peroxidase-conjugated secondary antibody (1:5000) for 1 h. The antibodies used were as follows: anti-Hsp70 (sc33575), from Santa Cruz Biotechnology and anti-PARP-1 (#9542), from Cell Signaling Technology. The immunocomplexes were visualized with the enhanced chemiluminescence (ECL) kit (Amersham, UK). The quantification of protein was performed by densitometric analysis of protein bands using ImageJ 1.42q Software.

### 4.12. Statistical Analysis

The results of the experiments described above were expressed as mean ± standard error. Statistical analyses were performed by Student’s *t*-test or analysis of variance (ANOVA) followed by Tukey’s post-test or by Dunnett’s post-test, using GraphPad Prism software for Windows, version 5.04 (GraphPad Software, San Diego, CA, USA). Results were considered statistically significant when *p* < 0.05.

## Figures and Tables

**Figure 1 molecules-26-00934-f001:**
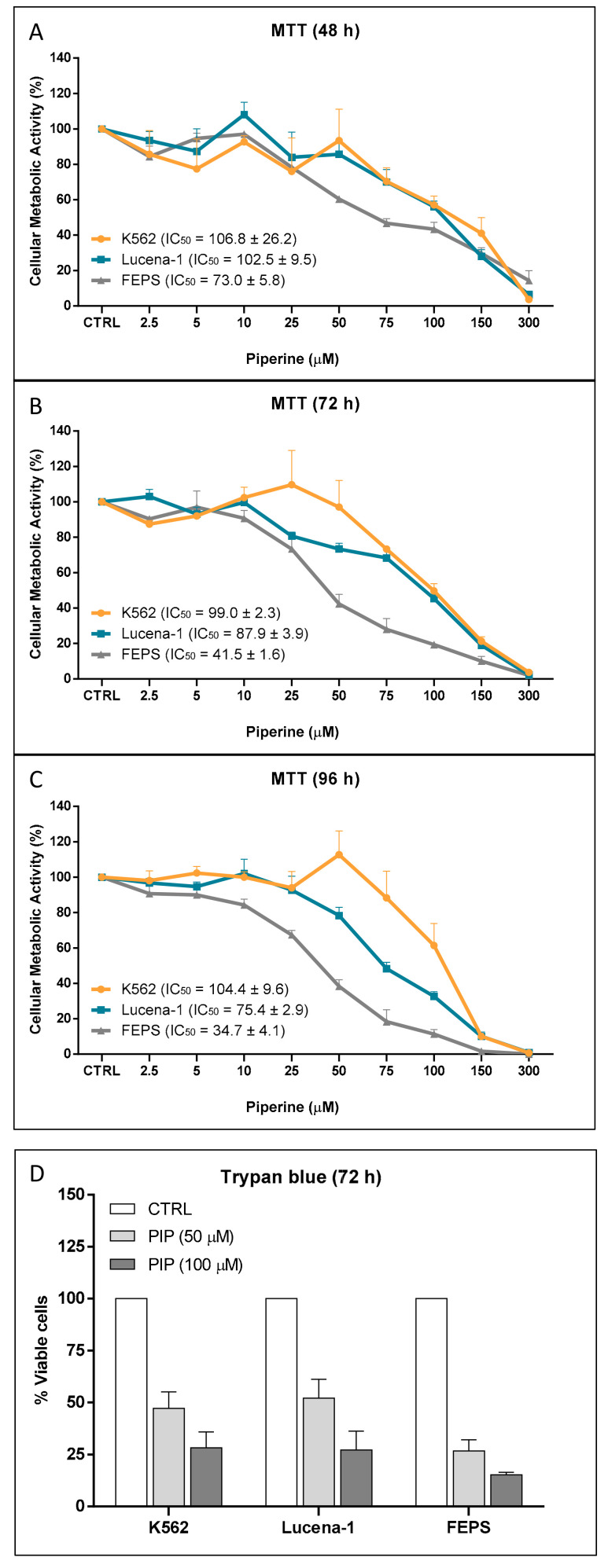
Effect of piperine on metabolic activity and the viability of leukemic cell lines. K562, Lucena-1 and FEPS cells were treated with different concentrations of piperine for 48 (**A**), 72 (**B**) and 96 h (**C**). Cell metabolic activity was determined by the MTT assay as described in the Materials and Methods section. The percentage of cell metabolic activity was calculated as the ratio of treated cells to control cells. (**D**) K562, Lucena-1 and FEPS cells were treated with 50 or 100 µM of piperine and cell viability was determined by trypan blue staining. Data represent the mean ± SE of at least 3 independent experiments. CTRL, control. PIP, piperine.

**Figure 2 molecules-26-00934-f002:**
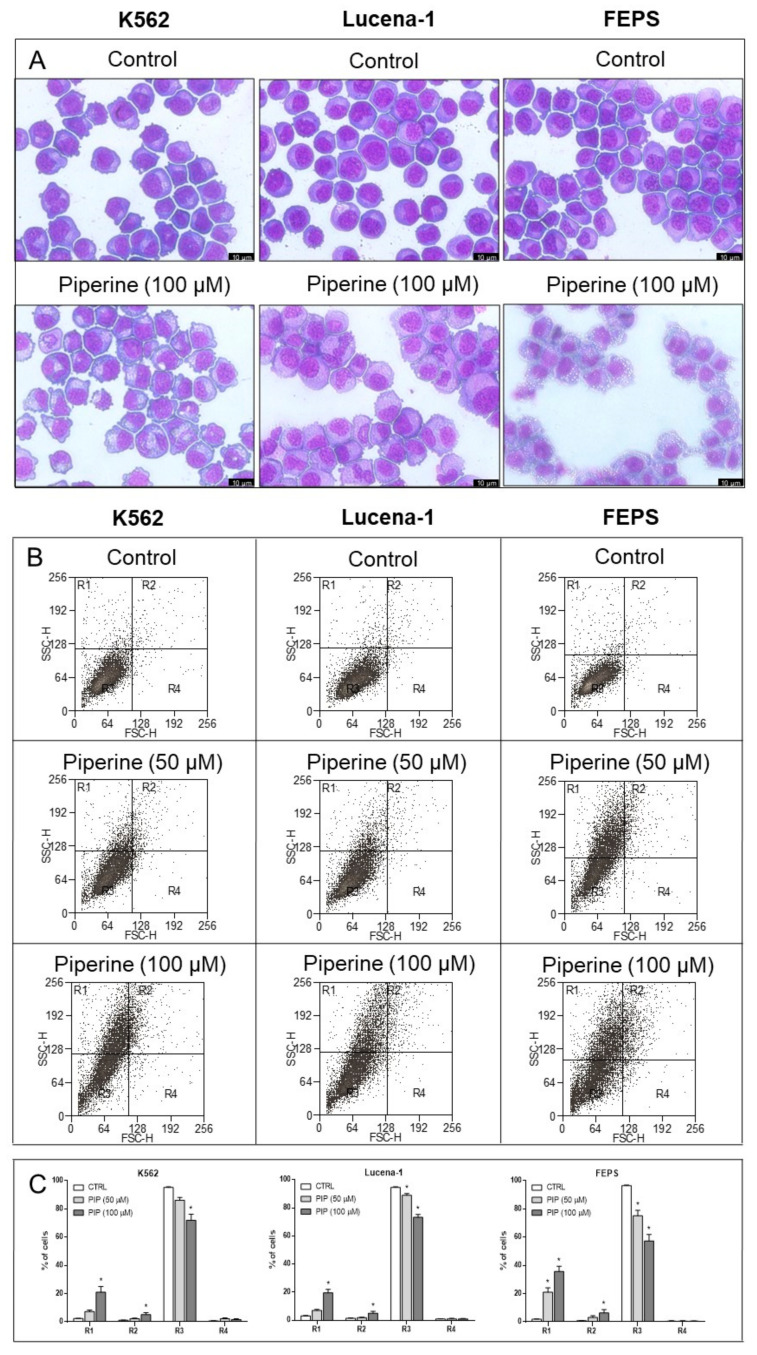
Effect of piperine on leukemic cells morphology. After 72 h of treatment, K562, Lucena-1 and FEPS cells were (**A**) stained with Quick Panoptic and observed under a bright field microscope (63× magnification) and were (**B**) analyzed by flow cytometry. (**C**) Bars represent mean ± standard error of events in the quadrant analyzed (R1, R2, R3 or R4). * *p* < 0.05 (ANOVA with Dunnett’s post-test) compared to DMSO control cells. The images are representative of 3 independent experiments. CTRL, control. PIP, piperine. FSC (Forward Scatter)—relative cell size. SSC (Side Scatter)—granularity or cellular complexity.

**Figure 3 molecules-26-00934-f003:**
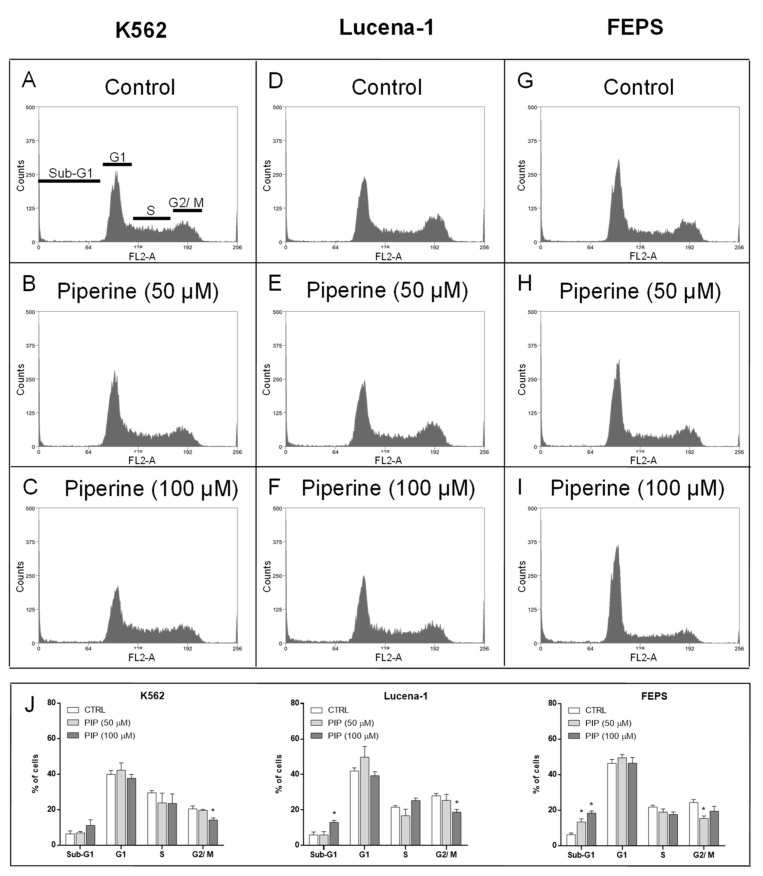
Effect of piperine on the cell cycle distribution of leukemic cell lines. After 72 h of treatment, K562, Lucena-1 and FEPS cells were stained with PI and their DNA content was determined by flow cytometry. K562 (**A**) control, (**B**) incubated with 50 μM piperine, and (**C**) incubated with 100 μM piperine. Lucena-1 (**D**) control, (**E**) incubated with 50 μM piperine, and (**F**) incubated with 100 μM piperine. FEPS (**G**) control, (**H**) incubated with 50 μM piperine, and (**I**) incubated with 100 μM piperine. (**J**) Bars represent mean ± standard error the percentage of cells in Sub-G1, G1, S or G2/M phase. * *p* < 0.05 (ANOVA with Dunnett’s post-test), compared to control cells with DMSO. The images are representative of at least 3 independent experiments. PI, propidium iodide; CTRL, control; PIP, piperine.

**Figure 4 molecules-26-00934-f004:**
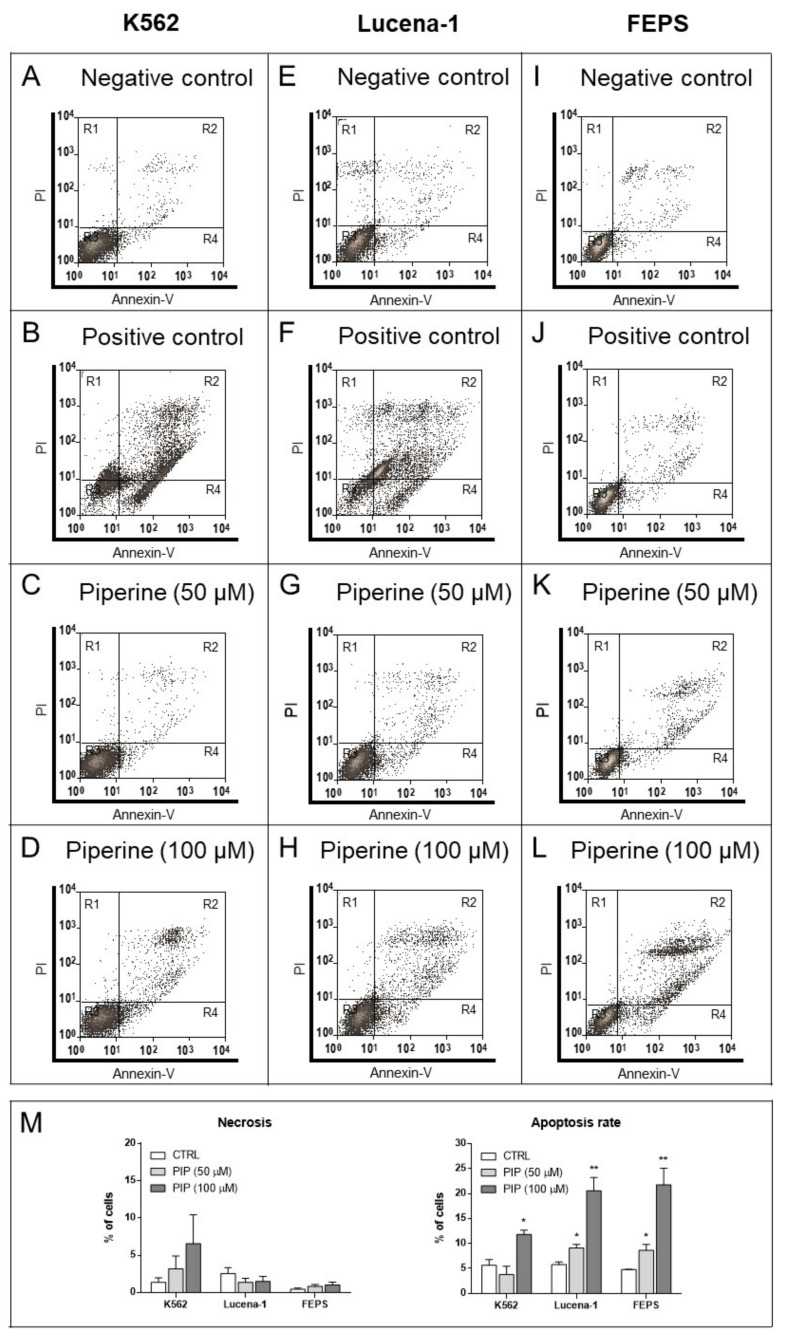
Effect of piperine on cell death of leukemic cell lines. After 72 h of treatment, K562, Lucena-1 and FEPS cells were labeled with annexin-V and PI and the result was analyzed by flow cytometry. K562 (**A**) negative control, (**B**) positive control (25 μM etoposide), (**C**) incubated with 50 μM piperine, and (**D**) incubated with 100 μM piperine. Lucena-1 (**E**), negative control (**F**), positive control (25 μM etoposide) (**G**), incubated with 50 μM piperine (**H**), and incubated with 100 μM piperine. FEPS (**I**) negative control (**J**) positive control (25 μM etoposide) (**K**) incubated with 50 μM piperine (**L**) incubated with 100 μM piperine. (**M**) Bars represent the percentage of cells undergoing necrosis or apoptosis. * *p* < 0.05 (Student’s *t*-test, unpaired), compared to control cells with DMSO; ** *p* < 0.01 (Student’s *t*-test, unpaired), compared to control cells with DMSO. The images are representative of 3 independent experiments. R1, necrosis; R2, late apoptosis; R3, alive; R4, early apoptosis; PI, propidium iodide; CTRL, control; PIP, piperine.

**Figure 5 molecules-26-00934-f005:**
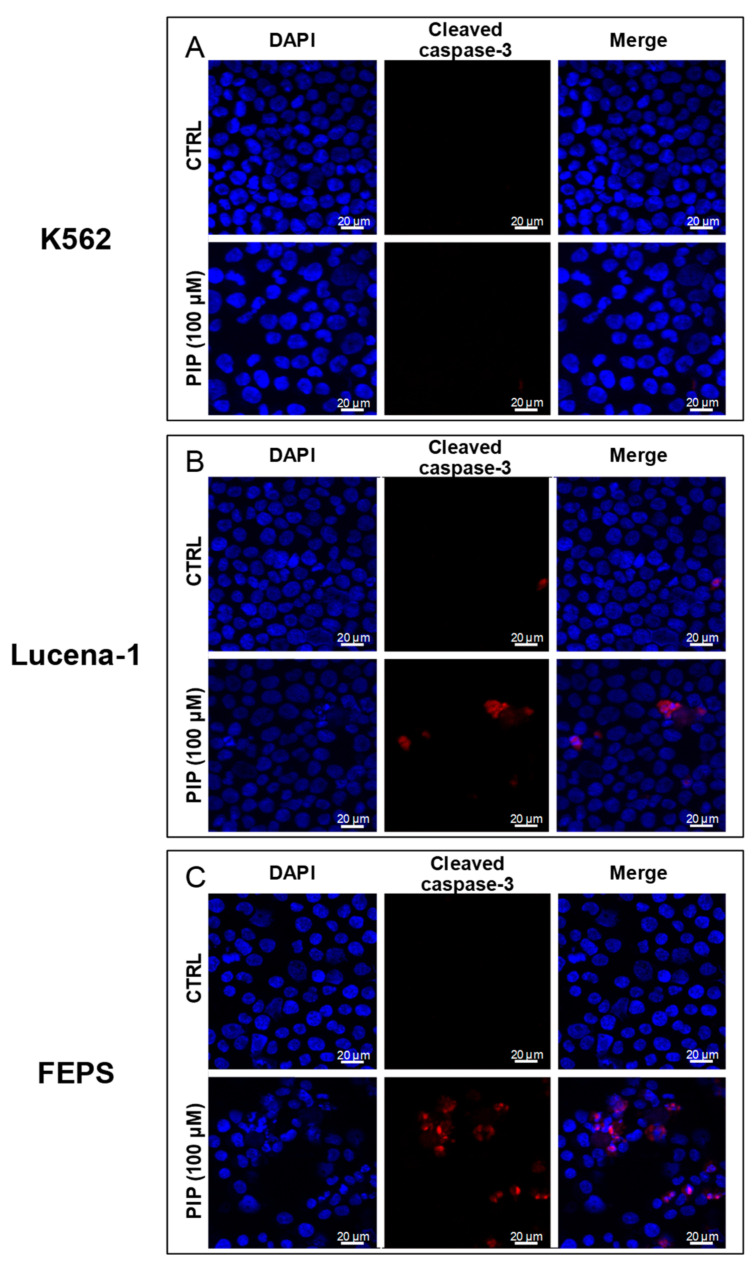
Effect of piperine on caspase-3 cleavage in leukemic cell lines. After 72 h of treatment, K562, Lucena-1 and FEPS cells were labeled with cleaved caspase-3 antibody and DAPI and analyzed under a fluorescence microscope. K562 (**A**), Lucena-1 (**B**), and FEPS cells (**C**) incubated with DMSO (CTRL) or 100 μM piperine. The images are representative of 3 independent experiments. CTRL, control. PIP, piperine.

**Figure 6 molecules-26-00934-f006:**
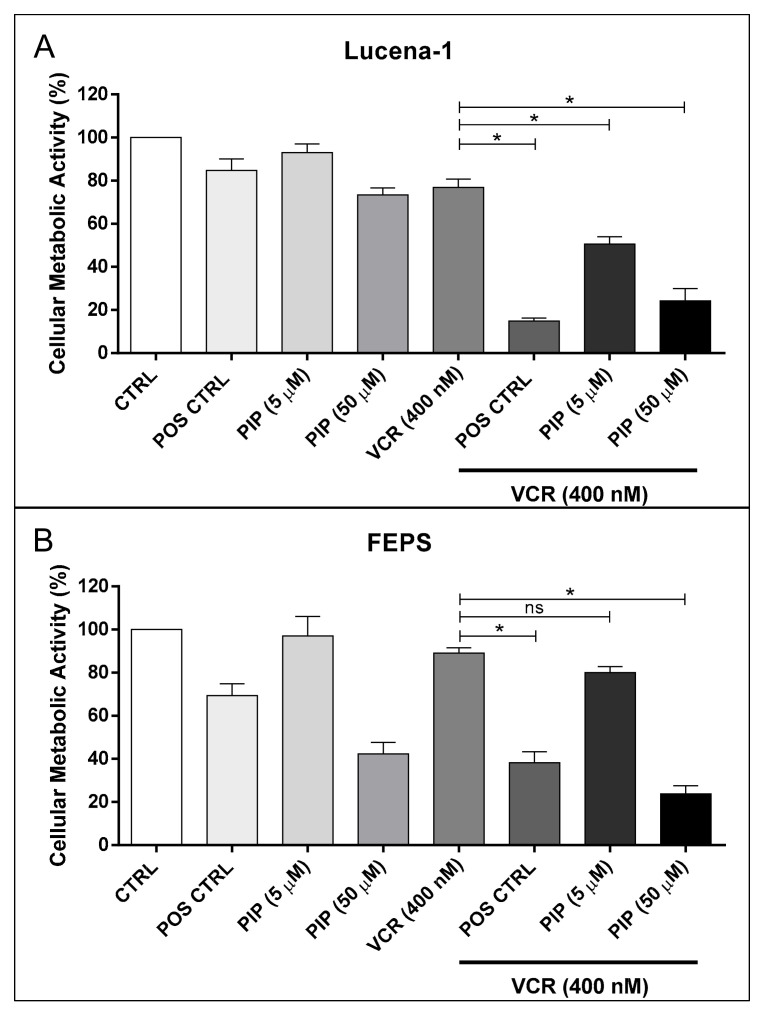
Effect of the association of piperine with vincristine on metabolic activity of multidrug resistance (MDR) cell lines. Lucena-1 (**A**) and FEPS cells (**B**) were treated, for 72 h, with 400 nM of vincristine in the presence or absence of piperine (5 μM or 50 μM) or verapamil (5 μM), used as positive control. Cell metabolic activity was determined by the MTT assay as described in the Materials and Methods Section. The percentage of cell metabolic activity was calculated as the ratio of treated cells to control cells. Data represent the mean ± SE of 3 independent experiments. * *p* < 0.05 (ANOVA with Tukey’s post-test), compared to control cells with DMSO. “ns” represents not significant. CTRL, control; POS CTRL, verapamil positive control; PIP, piperine; VCR, vincristine.

**Figure 7 molecules-26-00934-f007:**
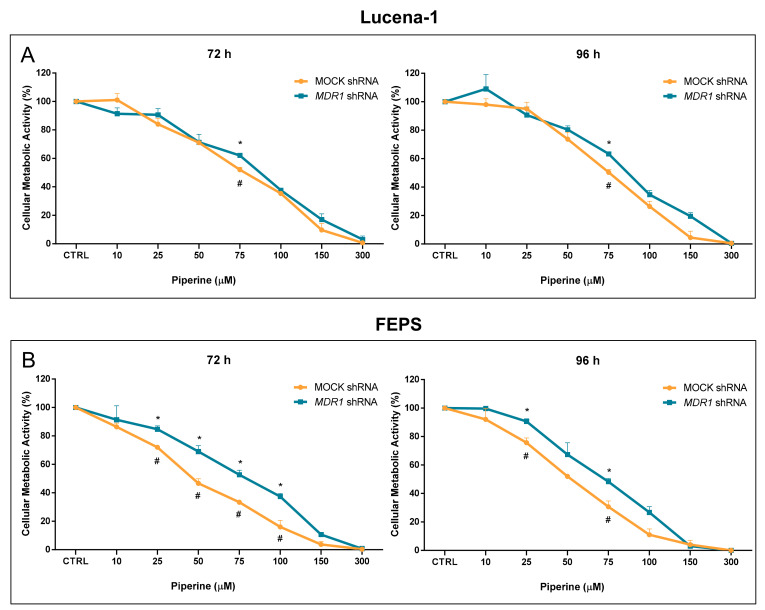
Effects of piperine on metabolic activity of MDR cell lines after *MDR1* silencing. Silenced Lucena-1 (**A**) and FEPS (**B**) cells were treated with different concentrations of piperine for 72 and 96 h. Cell metabolic activity was determined by the MTT assay as described in the Materials and Methods Section. The percentage of cell metabolic activity was calculated as the ratio of treated cells to control cells. Data represent the mean ± SE of 3 independent experiments. Different symbols (* or #) denote significant differences (*p* < 0.05, Student’s *t*-test, unpaired) between MOCK shRNA and *MDR1* shRNA cells.

**Figure 8 molecules-26-00934-f008:**
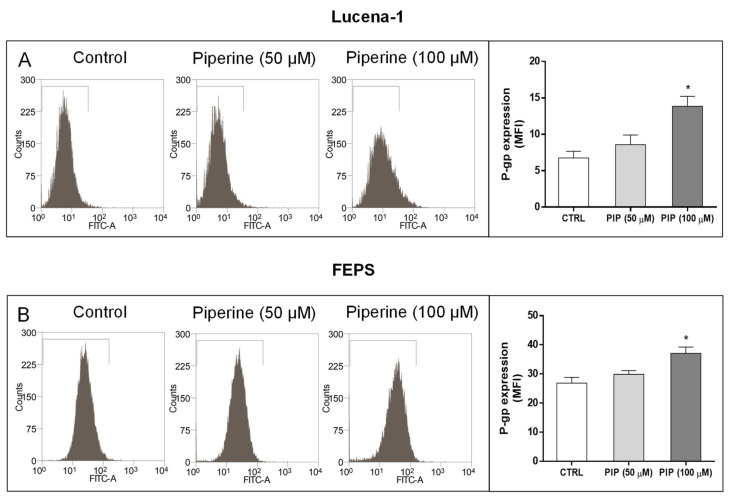
Effect of piperine on the P-gp expression of MDR cell lines. Lucena-1 (**A**) and FEPS (**B**) cells were treated with piperine (50 and 100 µM) for 72 h. Then, cells were incubated with the anti-P-gp antibody and the fluorescence was measured by flow cytometry as described in the Materials and Methods Section. Bars represent P-gp expression measured as mean fluorescence intensity (MFI). The images are representative of 3 independent experiments. Data represent the mean ± SE of 3 independent experiments. * *p* < 0.05 (ANOVA with Tukey’s post-test), compared to control cells with DMSO. CTRL, control; PIP, piperine.

**Figure 9 molecules-26-00934-f009:**
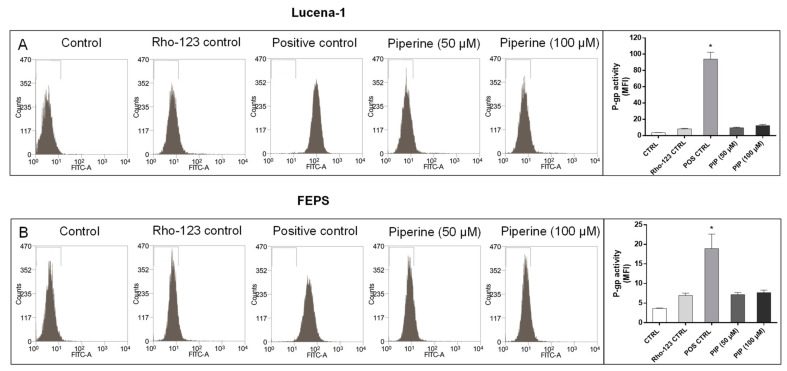
Effect of piperine on P-gp activity of MDR cell lines. Lucena-1 (**A**) and FEPS (**B**) cells were loaded with 200 ng/mL (Lucena-1) or 500 ng/mL (FEPS) of Rho-123 and left to extrude the dye in the presence or absence of piperine (50 and 100 µM) or verapamil (5 µM), as positive control, and the fluorescence was measured by flow cytometry as described in the Materials and Methods Section. Bars represent P-gp activity measured as mean fluorescence intensity (MFI). The images are representative of 3 independent experiments. Data represent the mean ± SE of 3 independent experiments. * *p* < 0.05 (ANOVA with Tukey’s post-test), compared to control cells with DMSO. CTRL, control; Rho-123, Rhodamine-123; POS CTRL, verapamil positive control; PIP, piperine.

**Figure 10 molecules-26-00934-f010:**
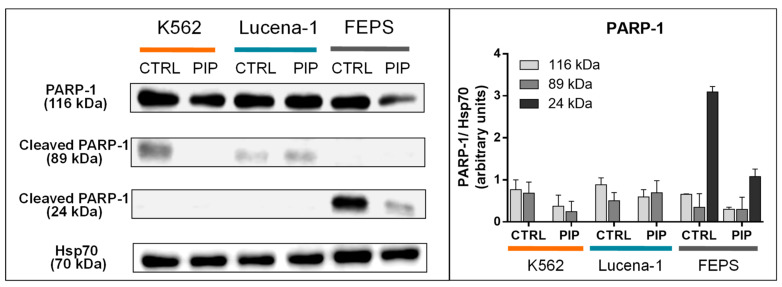
Effect of piperine on PARP-1 levels in leukemic cell lines. Equal amounts of total cellular proteins (100 µg) were loaded in each lane for the detection of PARP-1 and Hsp70 (loading control). The blots are representative of two different experiments that gave similar results. Bars represent the mean ± SE expressed as arbitrary units. Densitometric analysis of each lane was calculated using Image J Software. CTRL, control. PIP, piperine.

**Figure 11 molecules-26-00934-f011:**
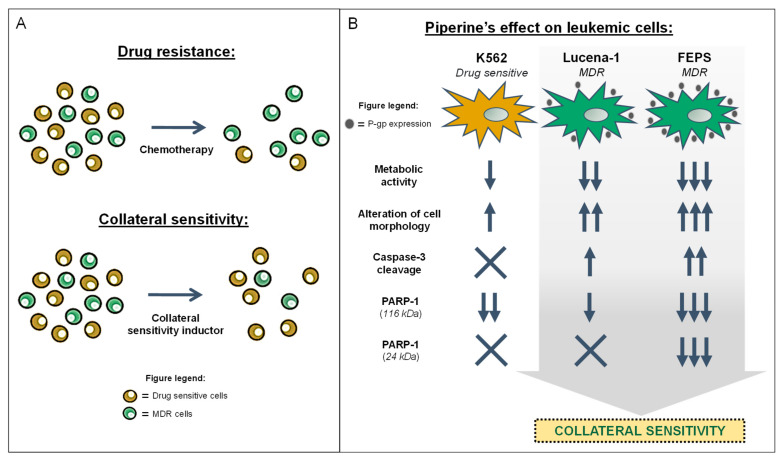
Collateral sensitivity and piperine’s effect on leukemic cells. (**A**) Most tumors have both MDR and drug-sensitive cells. Chemotherapy is less effective on MDR cells and only eliminates sensitive cells. Some substances, known as collateral sensitivity promoters, have the opposite effect and act selectively on MDR cells. (**B**) Piperine affected MDR leukemic cells, Lucena-1 and FEPS, in a more pronounced manner, compared to the drug-sensitive cell line, K562. Piperine reduced metabolic activity, altered cell morphology, and promoted caspase-3 cleavage. Piperine downregulated PARP-1 protein levels and its 24 kDa fragment in FEPS cells. Piperine’s effect was proportional to P-gp expression, being more effective on FEPS cells.

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
