# Peer review of "Selective Cytotoxicity of Piperine over Multidrug Resistance Leukemic Cells"

_molecules, 2021, doi:10.3390/molecules26040934_

Round 1

Reviewer 1 Report

The authors present and interesting manuscript investigating on the effect of piperine on collateral sensitivity induction in chronic myeloid leukemia multidrug resistance cells and partially mechanisms involved in.

Major comments:

  1. All figures except Figure 5 should be modified. They are not reader friendly since the reader needs to check figures and tables constantly. It is good to show a representative image for an experiment but in instances is difficult to appreciate the differences and the degree of the difference.
    1. Figure 1: Instead of bar graph I would suggest the authors use curves to illustrate the cell viability vs drug concentration and then a bar graph with the IC 50 data.
    2. Most relevant data on Table 2, 3 and 4 should be transformed to graphic and include in Figure 2, 3 and 4, respectively.
    3. Figure 6 should also be transformed to curves.
    4. Figures 7, 8 and 9 require a bar graph or similar to appreciate better the differences.
  2. The authors use mouse macrophages to evaluate cytotoxicity. Although of interest, the cell lines evaluated in this manuscript are human, thus human healthy control would be more recommended.
  3. In line with the previous comment, authors should strongly consider the evaluation on patients’ samples without and with multidrug resistance. Also, the combination with additional drugs and evaluate synergism.

Minor comments:

  1. Authors should consider exchange the MDR abbreviation in the title for the full “multidrug resistance”.
  2. How the authors can reconcile the dramatic differences observed with MTT studies but not with the other techniques? MTT is not a viability test is a metabolic test, if the drug is affecting somehow the metabolism the interpretation is wrong. Alternative authors could verify it by viable cell counting.
  3. Figure 3I does not look representative. Visually it is clearly showing an increase in G1 but data in Table 3 indicates that this should be similar to the data from Figure 3C since they present similar median and standard error.
  4. In line 78-79 is stated “have a slower cell cycle, so they divide less than K562”. But this is not observed in Figure/Table 3. How the authors explain this difference.
  5. In Figure 4 there are clear issues with Annexin V - PI compensation. Annexin V channel is clearly spilling over the PI channel. Thus, R4 region is under evaluated and R2 is over.
  6. Figure 6 should include a western blot or similar showing how MDR1 have become negative.
  7. Better description of the mechanism of action of piperine need to be addresses in the introduction and discussion.

Author Response

Reviewer 1
Comments and Suggestions for Authors
The authors present and interesting manuscript investigating on the effect of piperine on collateral sensitivity induction in chronic myeloid leukemia multidrug resistance cells and partially mechanisms involved in.
Major comments:
1. All figures except Figure 5 should be modified. They are not reader friendly since the reader needs to check figures and tables constantly. It is good to show a representative image for an experiment but in instances is difficult to appreciate the differences and the degree of the difference.
✓ Figure 1: Instead of bar graph I would suggest the authors use curves to illustrate the cell viability vs drug concentration and then a bar graph with the IC50 data.

Answer: We agree with the referee and the figure was remade in the form of curves, but instead of a bar graph with the MTT IC50 the values were included in the figure itself.

 Most relevant data on Table 2, 3 and 4 should be transformed to graphic and include in Figure 2, 3 and 4, respectively.

Answer: We agree with the referee and the requested changes were made. A graph in the form of a bar chart has been added to the figures and the legends modified to include the new additon.

✓ Figure 6 should also be transformed to curves.

Answer: We agree with the referee and the requested changes were made.
✓ Figures 7, 8 and 9 require a bar graph or similar to appreciate better the differences.

Answer: We agree with the referee and the requested changes were made. A graph in the form of a bar chart has been added to the figures and the legends modified to include the new additon.

2. The authors use mouse macrophages to evaluate cytotoxicity. Although of interest, the cell lines evaluated in this manuscript are human, thus human healthy control would be more recommended. In line with the previous comment, authors should strongly consider the evaluation on patients’ samples without and with multidrug resistance.

Answer: Other authors (mentioned in the Discussion Section) have used normal human cells with little toxicity. Due to the short amount of time given to us (encompassing Xmas and New Year) and the exceptional conditions of the moment (the COVID-19 pandemic) it would be impossible for us to obtain an agreement of the Ethics Committee to perform experiments with human cells. Similarly to obtain volunteers to donate their blood and to perform the experiments. As suggested by the referee we removed the information using mouse macrophages.

3. Also, the combination with additional drugs and evaluate synergism.

Answer: We added this experiment at the Results Section 2.4. Piperine partially reverses the resistance and shows synergism to vincristine on Lucena-1 cells.
Minor comments:

  1. Authors should consider exchange the MDR abbreviation in the title for the full “multidrug resistance”.
    Answer: To accomplish this suggestion the title was modified to “Selective cytotoxicity of piperine over multidrug resistance leukemic cells”2. How the authors can reconcile the dramatic differences observed with MTT studies but not with the other techniques? MTT is not a viability test is a metabolic test, if the drug is affecting somehow the metabolism the interpretation is wrong.
  2. Alternative authors could verify it by viable cell counting.
    Answer: We agree that MTT does not measure viability, but metabolic state. We modified, at MTT figures, the legeds of Y axis to: “Cellular Metabolic Activity (%)”. The fact that we are measuring metabolism is raised in the discussion. In addition, we included trypan blue cell viability experiments in the paper, and now is part of Figure 1. To accomplish this suggestion the following sentences were also modified and/ or included in the Discussion Section:
    - Lines 422-426: “Using the MTT assay, that measures the metabolic activity of the cell line, it was possible to observe that piperine toxicity differed among the MDR cells, being more expressive on FEPS compared to Lucena-1 cells. When a viability test was used, such as trypan blue exclusion, FEPS was also the cell line with the lowest percentage of living cells after treatment with piperine.”
  3. Figure 3I does not look representative. Visually it is clearly showing an increase in G1 but data in Table 3 indicates that this should be similar to the data from Figure 3C since they present similar median and standard error.
    Answer: We added new data and then we carry out an extensive total re-analysis. However, the percentage in the G1 phase of none of the cells has changed. The changes appear to be subtle, so there is no significant difference. In addition, we had previously observed a tendency to reduce the percentage of cells in the G2/ M phase, but, after re-analysis, these changes became clearer and more significant in all cell lines. And now, we also have a more striking difference in the sub-G1 phase of the FEPS cells treated with both 50 and 100 μM piperine. The following sentences were modified and/ or included in the Results Section due to changes in data analysis:
    - Lines 195-207: “The concentrations of piperine used, as well as the exposure time, were not sufficient to significantly arrest the cell cycle of the studied leukemic cell lines at some specific phase (Figure 3). In addition to the investigation of cell cycle, we analyzed DNA fragmentation, as an indicative of cell death. For this, the quantification of cells in the Sub-G1 phase was evaluated (Figure 3). Piperine treatment with 100 μM only increased Sub-G1 phase of the MDR cells, Lucena-1 and FEPS, indicating DNA fragmentation (≃ 15% and 20%, respectively). However, a lower concentration of piperine (50 μM) was already capable of rising the Sub-G1 phase of FEPS cells (≃ 15 %). Besides that, a modest decrease in the percentage of cells in G2/ M phase was also observed in the three studied cell lines (≃ 5-10 % reduction), after treatment with 50 (FEPS) or 100 μM piperine (K562 and Lucena-1), probably due to the increase (or trend to in-crease, in the case of K562) of cells in the Sub-G1 phase (Figure 3).”
  4. In line 78-79 is stated “have a slower cell cycle, so they divide less than K562”. But this is not observed in Figure/Table 3. How the authors explain this difference.
    Answer: We modified the setence and now it only reads “... they divide less than K562”. The two MDR cell lines divide more slowly than K562. This has been observed previously (Daflon-Yunes et al., 2013) (doi: 10.1007/s11010-013-1761-0) as a result of cell counts. In the present study there were indications that this is so: the total number of control cells after 72 hours culture (cell counts from 13 experiments): K562 = 24.91 × 104/mL; Lucena-1 = 19.56 × 104/mL; FEPS = 14.2 × 104/mL.
    The cell cycle shown in Figure 3 also indicates this trend. If one observes the percentage of control cells in the different phases of the cycle of the three lineages (Figure 3J average of five independent experiments), they vary depending on the cell line. K562 control cells have at 72 hours, 39.96 ± 2.1 cells in G1, 29.54 ± 1.2 cells in S, 20.49 ± 1.6 cells in G2/ M. Lucena-1 control cells have at 72 hours, 41.87 ± 1.8 cells in G1, 21.54 ± 0.9 cells in S, 27.94 ± 1.3 cells in G2/ M. FEPS control cells at 72 hour present 46.4 ± 21 cells in G1, 21.82 ± 0.9 cells in S, 24.34 ± 1.6 cells in G2/M.
  5. In Figure 4 there are clear issues with Annexin V - PI compensation. Annexin V channel is clearly spilling over the PI channel. Thus, R4 region is under evaluated and R2 is over.
    Answer: Thank you. We will try to correct it in the future. However, for the sake of indicating if there is apoptosis it does not modify the point of apoptosis or late apoptosis.
  6. Figure 6 should include a western blot or similar showing how MDR1 have become negative.
    Answer: We don't know if we made it clear enough, but, in the present study, we used Lucena-1 and FEPS cells silenced for the MDR1 gene expression that were previously established, by Daflon-Yunes et al. (2013) (doi: 10.1007/s11010-013-1761-0), using transfection with shRNA plasmids. So, the requested graph has already been published and we can’t use the same data published by other journal. However, we added information about the silencing efficiency in the Discussion Section:
    - Lines 483-485: “Therefore, we used silenced cells for the MDR1 gene. These cells were previously established in our laboratory and purified by cell sorting, with the silencing efficiency estimated at 80.76 % for Lucena-1 and 85.23 % for FEPS (Daflon-Yunes et al., 2013).”
  7. Better description of the mechanism of action of piperine need to be addresses in the introduction and discussion.
    Answer: Introduction have been modified. To accomplish this suggestion the following sentences were included in the Introduction Section:
    - Lines 101-111: “Piperine is able to modulate the MDR phenotype in some experimental models, such as breast, lung, colon and lymphoma cancer. This compound inhibited the
    gene expression and activity of the P-gp and BCRP efflux pumps (in breast cancer cells) and the MRP-1 efflux pump (in lung cancer cells) (Li et al., 2011). It has also been demonstrated the action of this phytochemical on colon cancer and lymphoma cells with MDR pheno-type, by enhancing the cytotoxic effect of the chemotherapeutic drugs doxorubicin and mitoxantrone, respectively. In addition to having inhibited P-gp activity in these two types of cancers (Li et al., 2018). A study by Morsy et al. (2018) found, through in silico and in vitro studies, that piperine was able to inhibit P-gp in ovarian cancer cells, suggesting that this phytochemical would be a promising adjuvant in the treatment with doxorubicin.”

Reviewer 2 Report

The work by Quarti et al. reports the possible effect of piperine in the specific mechanism of MDR thought p-glycoprotein involvements in leukemia cells.

The reported findings are innovative and can bring more light to the mechanism of action that could be shared by several phytochemicals.

However, I have few concerns regarding the concentration doses used in this study. In particular, I am referring to the meaningful amount that can reach the circulation, first, and the tumor, thereafter. I am opinion the concentrations used in the present study are way far for the concentration that can be reached after assuming a dose of black paper. By the way, how much?

Moreover, in the last decade and more, the range of acceptable concentrations in vitro (with very few exceptions) is in the nano-molar range.

Therefore, the authors need to discuss in deep this MAJOR issue, otherwise the significance and relevance of the study, that overall is well conducted, is missed.

Author Response

The work by Quarti et al. reports the possible effect of piperine in the specific mechanism of MDR thought p-glycoprotein involvements in leukemia cells. The reported findings are innovative and can bring more light to the mechanism of action that could be shared by several phytochemicals.
1. However, I have few concerns regarding the concentration doses used in this study. In particular, I am referring to the meaningful amount that can reach the circulation, first, and the tumor, thereafter. I am opinion the concentrations used in the present study are way far for the concentration that can be reached after assuming a dose of black paper. By the way, how much?

Answer: Clinical pharmacokinetic studies of piperine are scarce. Thereby, Wang et al. (2013) (doi: 10.1016/j.taap.2013.05.014) estimated the physiologically relevant level of piperine based on information from the literature and considering that an average daily consumption of black pepper is around 359 mg in the United States, but it is also a widely daily-used spice. These authors concluded that piperine might actually sustain, in the circulatory and digestive system, concentrations close to a theoretical range of ~20–44 μM. Besides that, a survey with eight healthy volunteers found that after taking two doses of 24 mg of piperine for one day, its serum concentration was around 6 μM (doi: 10.1111/j.1365-2125.2012.04364.x). Considering that in black pepper it would have around 9% of piperine, the volunteers of this study ingested the equivalent of 267 mg of black pepper twice a day. Therefore, there are indications that oral piperine supplementation or even the intake of black pepper itself would lead to micromolar concentration ranges of piperine in the blood. Furthermore, preliminary studies are encouraging and have shown to improve piperine’s bioavailability through nanoformulations and encapsulation in lipid bodies, as piperine is hydrophobic and exhibits low aqueous solubility, which is the major barrier for its use in the clinical set up (doi: 10.3109/10717544.2014.898109; doi:
10.1016/j.ejps.2019.104988) Another possibility is the use of piperine analogs in order to improve the efficiency of this compound, and, consequently, reduce its concentrations. For example, Syed et al. (2017) (doi: 10.1038/s41598-017-08062-2) synthesized piperine analogs and performed in silico experiments of the molecular interactions between P-gp and these analogs or piperine and also associated 2 μM of each compound with chemotherapeutic agents (vincristine, colchicine or paclitaxel) in drug-resistant cancer cells of cervical and colon. These authors demonstrated that the analogs had an even better effect than piperine, suggesting a promising effect on multidrug resistance cancer cells. Therefore, recent advancements in biotechnology area increasingly contribute to the possibility, in the near future, of using piperine or its analogs as a drug. The following sentences were also included in the Discussion Section:
- Lines 402-410: “Although clinical pharmacokinetic studies of piperine are scarce, there are indica-tions that oral piperine supplementation (Volak et al., 2013) or even the intake of black pepper itself (Wang et al., 2013) would lead to micromolar concentration ranges of piper-ine in the blood. Furthermore, preliminary studies are encouraging and have shown to improve piperine’s bioavailability through nanoformulations and encapsulation in lipid bodies (Shao et al., 2015; Ren et al., 2019). Another possibility is the use of piperine ana-logs in order to improve the efficiency of this compound, and, consequently, reduce its concentrations (Syed et al., 2017). Therefore, recent advancements increasingly contribute to the possibility, in the near future, of using piperine, or its analogs, as a drug.”

2. Moreover, in the last decade and more, the range of acceptable concentrations in vitro (with very few exceptions) is in the nano-molar range. Therefore, the authors need to discuss in deep this MAJOR issue, otherwise the significance and relevance of the study, that overall is well conducted, is missed.

Answer: Piperine has an important chemopreventive and chemotherapeutic action in vitro and in vivo. Most of the in vitro studies, that seek to evaluate piperine’s anticancer effect, as well as its mechanisms of action, use concentrations like those of the present study (50-200 μM) (doi: 10.1080/01635581.2017.1310260; doi: 10.3389/fcell.2018.00010; doi: 10.21873/anticanres.13296; doi: 10.17179/excli2018-1928). A recent review published by Manayi et al. (2018) (doi: 10.2174/0929867324666170523120656) also concluded that this evidence, on the anticancer effect of piperine, could support future clinical trials. To date, no clinical study with cancer patient has been published evaluating piperine safety or testing its anticancer effects. There is only one phase I clinical trial in progress that aims to evaluate, in cancer patients, the side effects and best dose of curcumin, phytochemical present in turmeric, combined with piperine in reducing inflammation for ureteral stent-induced symptoms (clinicalTrials.gov - Identifier NCT02598726). Therefore, clinical studies are needed to confirm piperine’s anticancer effect.

Round 2

Reviewer 1 Report

Thanks to the Authors for addressing my concerns and questions.

Unfortunately, the response to the minor point 6 is not acceptable. Compensation correction would take only few seconds plus additional minutes for reanalysis and figure preparation.

Finally, based on the new data presented in Figure 1. Authors should change “cytotoxic” in line 140-1. A suggestion would be “piperine was shown to impact  MDR”. As well as they should reconsider the sentence in line 440.

Reviewer 2 Report

The authors replied to all my concerns referring to pervious study on piperine effect within the range of doses used.

Even though, the same authors affirm “To date, no clinical study with cancer patient has been published evaluating piperine safety or testing its anticancer effects”.

In my opinion additional IN VITRO study on piperine effect is not highly significant for the scientific community until an human study will be conduced.

Referring to the in vitro study, I have nothing more to add.
